# Imaging Criteria for the Diagnosis of Progressive Supranuclear Palsy: Supportive or Mandatory?

**DOI:** 10.3390/diagnostics13111967

**Published:** 2023-06-05

**Authors:** Nicoleta Lupascu, Ioan Cristian Lupescu, Ionuț Caloianu, Florin Naftanaila, Remus Relu Glogojeanu, Carmen Adella Sirbu, Marian Mitrica

**Affiliations:** 1Department of Neurology, “Dr. Carol Davila” Central Military Emergency University Hospital, 010242 Bucharest, Romania; lupascunicoleta24@yahoo.com (N.L.); ionut.caloianu@gmail.com (I.C.); 2Clinical Neurosciences Department, University of Medicine and Pharmacy “Carol Davila” Bucharest, 050474 Bucharest, Romania; 3Department of Neurology, Fundeni Clinical Institute, 022328 Bucharest, Romania; 4Radiology and Medical Imaging Department, “Dr. Carol Davila” Central Military Emergency University Hospital, 010242 Bucharest, Romania; naftanaila_florin@yahoo.com; 5Department of Special Motricity and Medical Recovery, The National University of Physical Education and Sports, 060057 Bucharest, Romania; remus.glogojeanu@unefs.ro; 6Center for Cognitive Research in Neuropsychiatric Pathology (Neuropsy-Cog), Department of Neurology, Faculty of Medicine, “Victor Babeș” University of Medicine and Pharmacy, 300041 Timisoara, Romania; 7Department of Neurosurgery, “Dr. Carol Davila” Central Military Emergency University Hospital, 010242 Bucharest, Romania

**Keywords:** progressive supranuclear palsy, hummingbird sign, Mickey mouse sign, morning glory sign, midbrain atrophy, Magnetic Resonance Parkinsonism Index (MRPI), corticobasal degeneration, multiple system atrophy (MSA), PSP syndromes, postmortem histopathological examination

## Abstract

We present the case of a 54-year-old male, without any significant medical history, who insidiously developed speech disturbances and walking difficulties, accompanied by backward falls. The symptoms progressively worsened over time. The patient was initially diagnosed with Parkinson’s disease; however, he failed to respond to standard therapy with Levodopa. He came to our attention for worsening postural instability and binocular diplopia. A neurological exam was highly suggestive of a Parkinson-plus disease, most likely progressive supranuclear gaze palsy. Brain MRI was performed and revealed moderate midbrain atrophy with the characteristic “hummingbird” and “Mickey mouse” signs. An increased MR parkinsonism index was also noted. Based on all clinical and paraclinical data, a diagnosis of probable progressive supranuclear palsy was established. We review the main imaging features of this disease and their current role in diagnosis.

## 1. Introduction

Progressive supranuclear palsy (PSP) is a clinical-pathologic entity that was first described by Richardson, Steele, and Olszewksi in the 20th century [1]. It is the second most frequent form of neurodegenerative parkinsonism (after idiopathic Parkinson’s disease) [2]. The current diagnostic criteria for PSP were developed by the Movement Disorders Society in 2017 to increase the diagnostic sensitivity for the variant PSP syndromes (other than the classic presentation) [3]. They are based on (a) mandatory inclusion criteria and mandatory exclusion criteria, (b) involvement of four core functional domains (oculomotor dysfunction, akinesia, cognitive dysfunction, and postural instability), and (c) supportive features. When combined, these features give rise to one of three degrees of clinical diagnostic certainty: (1) suggestive of PSP, (2) possible PSP, and (3) probable PSP. There is also definite PSP, which can be currently diagnosed only by postmortem histopathological examination.

## 2. Case Presentation

We present the case of a 54-year-old Caucasian male, nonsmoker, nonalcohol consumer, with no prior medical history, who developed speech disturbance and walking difficulties (accompanied by frequent backward falls) of insidious onset with progressive worsening over 6 months. The patient was evaluated in 2019 by a neurologist and was initially diagnosed as having Parkinson’s disease. Brain MRI was also performed in 2019 but was unremarkable (see Figure 1A and Figure 2A). Standard therapy with oral Levodopa/Carbidopa 250/25 mg was initiated (and was titrated up to ¾ of a tablet q.i.d.); however, there was no improvement of symptoms. The patient was evaluated again in 2021 because of worsening postural instability with frequent falls, at which point the clinical exam highlighted vertical gaze palsy, hypomimia, urinary incontinence, bradykinesia, and rigidity, but no resting tremor. Because of this, the diagnosis was changed to progressive supranuclear palsy (PSP). The patient was admitted in July of 2022 at the Neurology Department of the Central Military Emergency University Hospital in Bucharest because of aggravated binocular diplopia. A neurological exam was highly suggestive of atypical parkinsonism, most likely progressive supranuclear gaze palsy (see Table 1). Additional neurologic signs noted were hypomimia, dysarthria, shortstepped gait with an anterior flexed posture of the trunk and neck, left-sided Babinski sign, and more pronounced DTRs on the left side. No other cranial nerve abnormalities were noted. The mental exam was unremarkable, and the patient scored 29/30 points on the Mini Mental State Examination (MMSE). An extensive laboratory work-up was performed for the differential diagnosis: thyroid hormones, vitamin B12, folate, vitamin D, inflammatory markers, autoimmune markers, infectious serologies (hepatitis B virus, hepatitis C virus, syphilis, HIV), and protein electrophoresis. Laboratory parameters were within normal limits, except for hyperuricemia and a low vitamin D level. Due to the presence of diplopia, pharmacologic testing with Neostigmine was performed and myasthenia gravis was excluded. The patient had no history of recent encephalitis, stroke, toxic exposure, or neuroleptic use and also no family history of neurologic diseases. Neurological exam excluded the presence of autonomic dysfunction, cortical sensory deficits, alien limb phenomena, cerebellar signs, and hallucinations. Based on the available diagnostic criteria for PSP, we established a diagnosis of probable PSP due to the presence of mandatory inclusion criteria, absence of mandatory exclusion criteria, and characteristic core features: ocular motor dysfunction and postural instability (O1 + P1). Brain MRI was repeated in 2022 and revealed moderate midbrain atrophy, which is considered to be a supportive feature for PSP diagnosis. The characteristic “hummingbird” sign was observed on sagittal T1 images (see Figure 1B). The “Mickey mouse” and “morning glory” signs were also present on axial T2 images (see Figure 2B), and other imaging markers suggestive of PSP were also noted, such as reduced midbrain-to-pons ratio and an increased MR Parkinsonism Index (see Figure 3). Unfortunately, nothing could be done apart from optimizing the antiparkinsonian treatment. The patient was discharged with Levodopa/Carbidopa/Entacapone at 150/37.5/200 mg q.i.d., Rasagiline at 1 mg q.d. and Rotigotine patch at 6 mg, and a neurorehabilitation program. Unfortunately, the patient has not been followed up since then.

## 3. Discussion

One thing to note is that the imaging features of PSP are currently considered supportive features and do not play a direct role in diagnosis. Brain imaging (and especially brain MRI) must be performed, however, for the exclusion of alternative diagnoses, such as normal pressure hydrocephalus, extensive small vessel disease, leukodystrophy, or mass lesions [4].

The role of computer tomography in PSP has been highlighted by some authors. For instance, Ambrosetto reviewed the CT scans of 87 patients with extrapyramidal disorders and/or dementia and found characteristic changes consisting of midbrain atrophy and quadrigeminal plate atrophy (with dilation of the aqueduct of Sylvius and third ventricle) only in the eight patients diagnosed clinically with PSP [5]. In another study comprising 17 patients with PSP, there was progressive atrophy of the pons and midbrain, with subsequent enlargement of the third and fourth ventricles and aqueduct as disease severity increased [6]. The authors of this study pointed out that the anterior–posterior diameters of the midbrain and pons were significantly smaller in patients than in controls and concluded that although the most obvious changes can be seen mainly in advanced disease, the diagnosis of PSP could be suggested by the CT scan before the classic clinical picture becomes evident. Saitoh et al. analyzed the head CT of six subjects with PSP and compared them with CT scans of Parkinson’s disease patients and healthy controls [7]. The patients with PSP had marked dilation of the lateral and third ventricles, as well as the prepontine cistern. Likewise, Yuki et al. compared the CT scans of four pathologically confirmed cases of PSP with those of 15 patients diagnosed with Parkinson’s disease and concluded that midbrain atrophy with dilation of the third ventricle and interpeduncular cistern was characteristic of PSP [8]. Last but not least, in the study performed by Massuci, all 10 patients diagnosed with PSP had increased midbrain atrophy with moderate atrophy of the pons, dilation of the third ventricle, and enlarged quadrigeminal plate cisterns. Additionally, six patients presented a hypodense abnormality extending from the interpeduncular cistern to the aqueduct (which was shown to represent an invagination of the interpeduncular cistern into the atrophic midbrain) [9]. It is worth mentioning that all of these papers are old (none of them more recent than 1990) and included small numbers of PSP patients. However, there was a recurring theme of midbrain atrophy with enlargement of the third ventricle.

MRI findings in PSP include marked atrophy of the dorsal midbrain, especially of its anterior–posterior diameter; increased signal of the midbrain on T2 images; dilation of the third ventricle; atrophy and increased T2/FLAIR signal of the superior cerebellar peduncles; atrophy or high signal of the red nucleus; increased signal of the globus pallidus; decreased T2 signal of the putamen (due to increased iron deposition); and frontal and parietal lobe atrophy [10,11]. On sagittal T1-weighted images, there is a reduction of the anteroposterior midbrain diameter, and on T2-weighted images, there is diffuse high signal involving the tegmentum and tectum of the midbrain, upper pontine tegmentum, and, in some patients, lower pontine tegmentum [12]. In an interesting paper by Warmuth-Metz et al., the anterior–posterior diameter of the midbrain on axial T2 images was measured and compared between patients diagnosed with Parkinson’s disease (PD), with PSP, and with multiple system atrophy (MSA) of striatonigral type and healthy age-matched controls. They concluded that midbrain diameters were significantly lower in PSP patients than in patients with PD or healthy controls; however, midbrain diameters were also lower in MSA patients and overlapped with those of PSP [13]. 

The selective atrophy of the dorsal midbrain on midsagittal images has been termed the “hummingbird” sign or “penguin silhouette” sign, where the superior midbrain margin becomes flat or concave (as opposed to convex in healthy subjects) (see Figure 1) [1,11]. Righini et al. concluded that the presence of a flat or concave superior midbrain profile may help in distinguishing between PSP and idiopathic PD [14]. The sign was associated in their paper with 68% sensitivity and 88.8% specificity, whereas midbrain atrophy had a sensitivity of 68% and a specificity of 77.7%.

The “morning glory” sign refers to a specific pattern of midbrain atrophy, which leads to a concave aspect of the lateral margins of the midbrain tegmentum on axial images. This sign was found in four out of five patients diagnosed with PSP, but in only 1 out of 37 patients diagnosed with MSA or PD, and was shown to correlate with the presence of vertical supranuclear gaze palsy [15].

By analyzing in a blinded fashion the MRIs of 48 postmortem cases (pathologically confirmed as having PSP, MSA, Parkinson disease, or corticobasal degeneration or controls), Massey et al. established that the “hummingbird” and “morning glory” signs were indeed highly specific for PSP but had a low sensitivity. Moreover, MRI assessment, clinical diagnosis and macroscopic postmortem examination all had similar sensitivity and specificity in predicting the correct neuropathological diagnosis [16].

The “Mickey Mouse” sign, which is also seen on axial images, is determined by the reduction of the anterior–posterior diameter of the midbrain and is associated with the thinning of the cerebral peduncles, thus giving rise to the “face of Mickey Mouse” (see Figure 2) [17,18].

The problem with all these signs, however, is that they are based on the mere visual assessment of the midbrain. Some authors have expressed concern that this is not an objective measure (since it is not quantitative) and is also dependent on the image acquisition parameters and positioning of the patient [19,20,21]. This was highlighted by Adachi et al. In their example, thinly sliced images (of 3 mm thickness) revealed the “morning glory” sign, whereas conventional images (of 7 mm thickness) did not [22].

One can also measure the (a) midbrain area and (b) midbrain-area-to-pontine-area ratio, both of which should be decreased in PSP. Oba et al. performed these measurements in patients with PD, with MSA and parkinsonism (MSA-P), or with PSP and healthy controls. They concluded that PSP patients presented a significantly lower midbrain area (56 mm^2^) as compared to PD patients (103 mm^2^), MSA-P patients (97.2 mm^2^), and healthy controls (117.7 mm^2^); however, some overlap was noted on an individual level between MSA-P and PSP patients. The ratio between the midbrain area and pons area was also significantly lower in PSP patients (0.124) than in patients with PD (0.208), MSA-P (0.266), and healthy subjects (0.237) and allowed differentiation between PSP and MSA-P patients [23].

Other useful parameters appear to be the measurement of the (a) midbrain width and (b) midbrain-width-to-pons-width ratio. Massey et al. established cu-off values of the midbrain width and midbrain-to-pons ratio on midsagittal T1 sequences in histologically confirmed cases of PSP, after which they applied these cutoff values in a cohort of clinically diagnosed patients with PSP, PD, or MSA, and healthy controls. They showed that a midbrain width smaller than 9.35 mm and a midbrain-to-pons ratio lower than 0.52 had 100% specificity for PSP. A midbrain measurement under 9.35 mm was also seen in approximately 90% of the clinically defined cases of PSP [24]. In our patient, the midbrain-to-pons ratio was 0.56 in 2019 but decreased to 0.37 in 2022.

The MR parkinsonism index is a tool developed by Quattrone et al. for differentiating between PSP on the one hand, and idiopathic PD and MSA on the other. The index is calculated by taking into account (1) the ratio between the pons area and midbrain area (pons -area-to-midbrain-area ratio) and (2) the ratio between the superior cerebellar peduncle width and middle cerebellar peduncle width. Both individual ratios present some overlap between patients with PSP and patients with other parkinsonian syndromes, or even with healthy controls. However, when multiplied, they give rise to the MR Parkinsonism Index (MRPI), which is significantly larger in PSP patients than in other groups, without any overlap between the groups [25]. The use of the MRPI in distinguishing between PSP and probable or possible PD has been validated by other authors, who noted a 100% sensitivity and an over 99% specificity [26]. Additionally, MRPI has been used by Morelli et al. in predicting the clinical course of patients with clinically unclassifiable parkinsonism (that is, parkinsonism not fulfilling the established diagnostic criteria for any parkinsonian disorder). In their study, none of the 30 patients with a normal MRPI at baseline progressed to a clinical diagnosis of probable or possible PSP, as opposed to 11 of the 15 patients with an abnormal MRPI at baseline (i.e., higher than 13.55) [27]. In our patient, the MRPI value was higher than 14 and was thus consistent with a diagnosis of PSP. In advanced cases, the differential diagnosis between PSP and PD relies on the clinical picture. Advanced PD is diagnosed based on a set of strict clinical criteria and by judging the response to Levodopa and other treatments [28].

Of course, one must not forget that PSP encompasses different clinical phenotypes and not just the classical presentation (PSP-RS). Picillo et al. confirmed that the MRPI is useful in supporting the clinical diagnosis of classic PSP, but it does not help in differentiating variant PSP, especially PSP with predominant parkinsonism (PSP-P) from PD [29]. Quattrone et al. have addressed this issue by developing another imaging biomarker termed MRPI 2.0, which is calculated by multiplying the MRPI with the third ventricle-width-to-frontal-horns-width ratio. The MRPI 2.0 has shown a significantly higher sensitivity (100%) than has the original MRPI (73.5%) in distinguishing between PSP-P and PD patients, with a slightly lower specificity (94.3% for MRPI 2.0 versus 98.1% for MRPI). MRPI 2.0 was also more accurate than was the original MRPI in discriminating between PD patients and PSP patients in the early stages of disease (with slowness of vertical saccades) [30].

A subsequent paper by Quattrone et al. established that MRPI 2.0 could predict which patients with an initial diagnosis of PD would eventually develop a PSP-P phenotype (after a 4-year follow-up) [31].

Recently, a group of Italian researchers advocated for the widespread use of an automated algorithm for the calculation of the MRPI, as it was highly accurate and similar to the manual measurement method in their study (on both a 1.5 and 3 Tesla MRI) [32].

Some differences have also been highlighted between the PSP clinical phenotypes. For example, the PSP subcortical variants (PSP-P and PSP with progressive freezing of gait) could be distinguished in one study from PSP-RS and PSP cortical variants (PSP-frontal and PSP-CBS) by cortical volumetric MRI assessment [33]. Other authors have pointed out that brainstem measurements can be normal in PSP with progressive freezing of gait, whereas they are most abnormal in PSP-RS and PSP-F, followed by PSP-CBS, PSP-speech/language, and PSP-P. The same authors noted that brainstem measurements were more abnormal in patients with confirmed PSP pathology than in those with other pathologies and calculated a sensitivity of 83% and specificity of 85% for the MRPI [34].

Whitwell et al. observed frontal lobe atrophy in PSP-F, PSP-CBS, and PSP-speech/language, while atrophy of the superior cerebellar peduncle was identified only in PSP-P, PSP-F, and PSP-CBS. Additionally, the volume loss in PSP-P and PSP-freezing of gait was mainly limited to the basal ganglia, thalamus, and subthalamic nucleus. It should be noted, however, that all PSP variants presented atrophy of the basal ganglia and thalamus [35].

One limitation encountered in our study is the fact that a diagnosis of “definite PSP” can only be established by postmortem examination. At first, imaging criteria should be assessed retrospectively in pathologically confirmed cases in order to determine their real sensitivity and specificity. Imaging criteria could then be applied to cases of “clinically probable PSP” to further support the diagnosis of PSP antemortem. Another limitation regarding our case presentation is the fact that the patient was not followed up after discharge through clinical and paraclinical workup. A review of our patient’s clinical and radiological characteristics as compared to the main features highlighted in some of these studies is given in Table 2. 

## 4. Conclusions

Imaging features are currently considered only ancillary criteria for the clinical diagnosis of PSP, but their relevance and importance might increase in the future. The MRPI and its improved version MRPI 2.0 are the most promising tools in differentiating patients with PSP from those with other parkinsonian syndromes. More studies should be performed in this direction since the inclusion of imaging criteria as core diagnostic features would greatly facilitate the diagnosis of PSP and would open up the possibility of follow-up imaging and monitoring of disease progression.

## Figures and Tables

**Figure 1 diagnostics-13-01967-f001:**
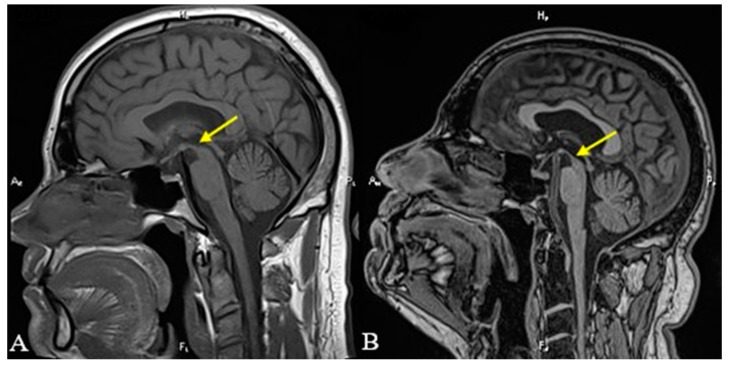
T1-weighted sagittal MRI revealing the superior margin of the midbrain (yellow arrows). Notice the convex superior margin of the midbrain that could be seen in 2019 (**A**) and the concave appearance in 2022 (**B**), the so-called “hummingbird” sign.

**Figure 2 diagnostics-13-01967-f002:**
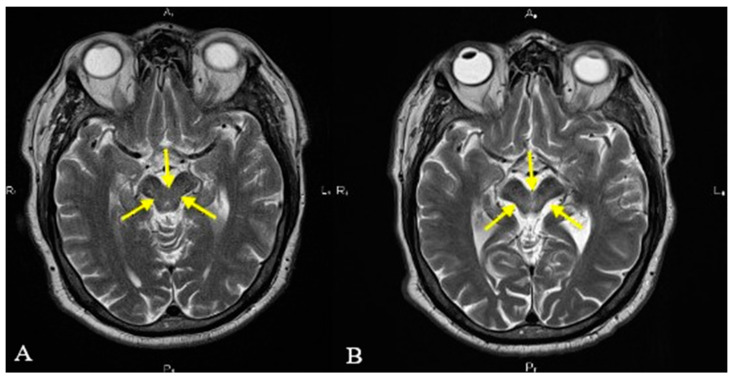
T2-weighted axial MRI revealing the midbrain margins (yellow arrows). Notice the normal-appearing midbrain that could be seen in 2019 (**A**) and the atrophic midbrain with concave lateral margins (“morning glory” sign) and reduced anteroposterior diameter (“Mickey mouse” sign) which was seen in 2022 (**B**).

**Figure 3 diagnostics-13-01967-f003:**
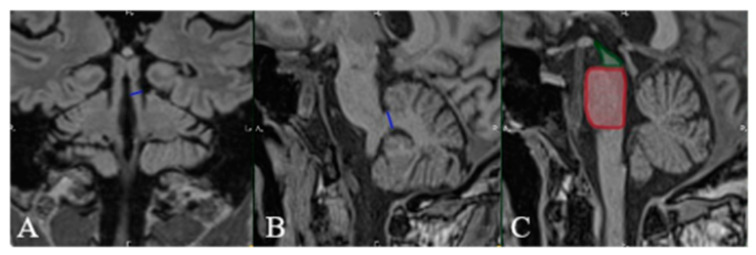
MRI FLAIR in coronal (**A**) and sagittal (**B**,**C**) planes. MR Parkinsonism Index (MRPI) in our patient is calculated by determining the: (**A**) width of the superior cerebellar peduncle (SCP) (blue line), (**B**) width of the middle cerebellar peduncle (MCP) (blue line), and (**C**) pons-to-midbrain area ratio (P/M) (red square/green square). MRPI = (P/M) × (MCP/SCP). A value higher than 13.55 is in-dicative of PSP (as was the case with our patient).

**Table 1 diagnostics-13-01967-t001:** Neurological findings in our patient (suggesting probable PSP based on the PSP clinical criteria).

**Ocular motor dysfunction**	Vertical gaze palsy (both upward and downward)—**O1**
	Limited adduction of the right eye
	Preserved vestibulo-ocular reflex
**Postural instability**	Absent postural reflex
	Repeated unprovoked falls within three years—**P1**
**Akinesia**	Bradykinesia
	Bilateral and axial rigidity
	Unresponsive to Levodopa—**A2**
**Cognitive dysfunction**	MMSE = 29/30 points (no cognitive impairment)

**Table 2 diagnostics-13-01967-t002:** Clinical and imaging findings in our patient as compared to the main features highlighted in other studies.

Author, Year	Age at Presentation	Duration of Disease	Clinical Characteristics	Radiological Findings
Our case report	54 years old	3 years	- Vertical gaze palsy (both upward and downward) - Repeated unprovoked falls (mainly backward) - Axial rigidity and bradykinesia - Unresponsive to Levodopa	Positive “hummingbird” sign.
				Positive “Mickey mouse” sign.
				Positive “horning glory” sign.
				MRPI > 14.
				Midbrain-to-pons ratio = 0.37.
Righini et al., 2004 [14]	68.88 ± 6.48 years old	5.3 ± 3.28 years	- 25 consecutive patients which fulfilled the diagnostic criteria for probable PSP proposed by Litvan et al. (1996) [36].	“Hummingbird” sign had 68% sensitivity and 88.8% specificity in discriminating between PSP and PD.
Sonthalia et al., 2012 [18]	75 years old	2 years	- Vertical gaze palsy - Axial rigidity and bradykinesia - Tendency to fall backward	- Positive “hummingbird” sign. - Positive “Mickey mouse” sign.
Adachi et al., 2007 [22]	57 years-old	2 years	- Vertical gaze palsy - Postural instability with falls - Unresponsive to Levodopa - Symmetrical rigidity and bradykinesia	“Morning glory” sign was present on slices of 3 mm thickness (but not on conventional slices of 7 mm thickness).
Massey et al., 2013 [24]	69.4 ± 6.5 years old	4.6 ± 3.1 years	- 29 pathologically confirmed cases of PSP, PD, or MSA and controls. - 62 clinically diagnosed patients with PSP, PD, or MSA and controls.	Midbrain-to-pons ratio < 0.52 had a 100% specificity for PSP.
Picillo et al., 2020 [29]	70 (52–84) years old	4 (1–11) years	- Clinical diagnosis of PSP based on the 2017 MDS-PSP criteria.	- MRPI: 17.84 (10.35–36.63) for PSP-RS. - MRPI: 13.75 (7.63–23) for PSP-P.
Quattrone et al., 2019 [31]	- 69 ± 5.9 years old (at baseline) - 73 ± 5.9 years old (after 4 years)	- 6.4 ± 2.3 years - 10.4 ± 2.3 years	- Initial diagnosis of PD. - Diagnosed with PSP after 4 years of follow-up.	- MRPI at baseline: 10.81 ± 0.8. - MRPI after 4 years: 15.06 ± 1.4.
Jabbari et al., 2020 [33]	70.6 ± 7.3 years old	4.4 ± 2.7 years	- 101 cases which fulfilled the 2017 MSD PSP criteria.	- Midbrain atrophy was a consistent feature in all PSP patients.
Whitwell et al., 2020 [35]	70 [66, 74] years old	6.6 [4.8, 8.5] years	- Clinical diagnosis of PSP-P (based on the 2017 MDS-PSP criteria).	- PSP-RS showed greater midbrain atrophy compared to PSP-P.
	68 [65, 72] years old	3.4 [2, 4] years	- Clinical diagnosis of PSP-RS (based on the 2017 MDS-PSP criteria).	

## Data Availability

Not applicable.

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
