# Peer review of "Imaging Criteria for the Diagnosis of Progressive Supranuclear Palsy: Supportive or Mandatory?"

_diagnostics, 2023, doi:10.3390/diagnostics13111967_

Round 1

Reviewer 1 Report (Previous Reviewer 2)

Firstly, I would like to congratulate the authors since their work has significantly improved after the first submission. However, some points need to be addressed, some already pointed out previously: 1) It is advisable that the authors include the study design type within the title. 2) The authors should work with English editing services to improve the readability of the text. Additionally, some sentences were repeated for example “They pointed out” on page 3 and in page 4 again. 3) The legend of figure 3 is too extensive and difficult to read. Perhaps the authors could add the neuroimaging details of their patient to a comparative table since it would improve readability and facilitate analysis of the article by the reader. 4) Regarding the literature review on the neuroimaging of PSP, I advise the authors to build a comparative table (similar to the one built in this article, but more detailed and analytical, including their patient’s clinical characteristics and the main characteristics of the patients from the studies they included in their review to improve the readability of the text). 5) The discussion should be significantly improved and also include the limitations of the study 6) Was the patient followed up after discharge? How is he now? If he was not followed up this should be disclosed as a limitation of the study.

Language editing required.

Author Response

Thank you for your time and suggestions!

1) It is advisable that the authors include the study design type within the title.

                   We have included the study design within the title (case presentation and literature review).

2) The authors should work with English editing services to improve the readability of the text. Additionally, some sentences were repeated for example “They pointed out” on page 3 and in page 4 again.

                   We reviewed and improved the readability of the text, and we replaced the sentences that were repeated.

3) The legend of figure 3 is too extensive and difficult to read. Perhaps the authors could add the neuroimaging details of their patient to a comparative table since it would improve readability and facilitate analysis of the article by the reader.

    We modified the legend of Figure 3 as requested, so that it is shorter, more readable and concise.

4) Regarding the literature review on the neuroimaging of PSP, I advise the authors to build a comparative table (similar to the one built in this article, but more detailed and analytical, including their patient’s clinical characteristics and the main characteristics of the patients from the studies they included in their review to improve the readability of the text).

    The patients included in the studies that were mentioned in the Discussions section were all diagnosed clinically with “probable” PSP, PD and MSA, as was our patient. Moreover, we don’t have much information regarding the patient’s clinical characteristics from these studies, since they were focused on the imaging aspects of the disease.

5) The discussion should be significantly improved and also include the limitations of the study

    We have included the limitations of the study in the Discussions section.

    “One limitation encountered in our study is the fact that a diagnosis of “definite PSP” can only be established by post-mortem examination. At first, imaging criteria should be assessed retrospectively in pathologically confirmed cases, in order to determine their real sensitivity and specificity. Imaging criteria could then be applied to cases of “clinically probable PSP” to further support the diagnosis of PSP antemortem. Another limitation, regarding our case presentation, is the fact that the patient was not followed-up after discharge through clinical and paraclinical workup.”

6) Was the patient followed up after discharge? How is he now? If he was not followed up this should be disclosed as a limitation of the study.

    The patient was not followed-up after discharge and we have added this information in the text.

Reviewer 2 Report (Previous Reviewer 1)

The authors supplemented the manuscript, which I had no comments on during the original assessment. I stated that the manuscript brings a new perspective, but I expressed concern that only a case study and a small scope is sufficient for publication. This concern of mine remains. The authors added a lot of information, but this does not increase the scope and importance of the entire work in any way. As I have already mentioned, the work is interesting and beneficial and can certainly be beneficial for readers and for practice.

Minor editing is recommended.

Author Response

Thank you for your time and suggestions!

Round 2

Reviewer 1 Report (Previous Reviewer 2)

I will reinforce some previous comments because I believe if they were addressed, the paper would improve substantially. I would like to give a few more suggestions regarding the references cited by the study also. Most of the references are older than five years. I suggest the authors update those references, find new articles from the last five years preferably, with clinical and radiological characteristics, and build a comparative table (more detailed and analytical, including their patient’s clinical characteristics and the main characteristics of the patients from the studies they included in their review to improve the readability of the text). With these new references, the discussion should also be improved and the authors should compare their findings to those findings from recent studies on the subject.

Minor editing required.

Author Response

This manuscript is a resubmission of an earlier submission. The following is a list of the peer review reports and author responses from that submission.

Round 1

Reviewer 1 Report

The manuscript is processed as an "interesting imagine" and is therefore completely different from standard articles, or has a completely different structure.

The authors present one case report in which they systematically describe all the patient's symptoms, history and, above all, present and carefully describe the MRI findings. The presentation of the case report as well as the description of the MRI seems to me to have been well prepared, perhaps the only thing that can be criticized is its brevity. This case report is certainly a contribution to practice and therefore deserves to be published.

Author Response

Thank you for your comments and for your support. We submitted our article at, the  ,interesting images, section, but we presented this case in full to support better case images.

Reviewer 2 Report

  Firstly, I would like to congratulate the authors on their relevant and outstanding work.  

However, there are some points that need to be addressed:
1) In the title, it is advisable that the authors include the study design type.

2) It would be interesting if the authors could further describe the setting in which
the patient was seen by them. Was it an outpatient movement disorders clinic?
Public or private? Was it an academic setting?

3) Could the authors provide a video of the neurological exam of the patient,
perhaps labeling the PSP main signs?

4) On page 2 the authors mention A brain MRI was also performed in 2019 but
was unremarkable (see Figures 1A and 2A).However, figures 1 A and 2 A are
labeled with the “hummingbird” sign, “morning glory” sign, and “Mickey
mouse” sign. Was this perhaps a figure legend error? Can the authors provide
the MRI images from 2019?

5) Can the authors further detail the initial pharmacological management of the
patient? Were any drugs tried before levodopa or with levodopa? Which were
the dosages, the levodopa regimen, and which type of formulation?

6) On page 2 the authors state: The patient repeated the brain MRI, which was
again unremarkable, apart from non-specific T2-high signal demyelinating
lesions involving the periventricular and subcortical white matter. “ Can the
MRI images be added to the paper?

7) Could the authors provide more details of the timeframe of the symptoms and
add months to their description? For example, in which month of 2022 was the
patient seen by them?

8) The authors mention in page 2 who developed speech disturbance and walking
difficulties (accompanied by frequent backward falls), with insidious onset.
Could the authors further clarify “insidious onset”? How long was that?

9) It is important that the authors give further details on the patient’s clinical
history and physical exam. Could the authors further describe aspects such as
speech, mental exam, bulbar exam, oculomotor exam, limb and gait exam, if the
neurological symptoms were evaluated through any standardized scales besides
MMSE.

10) The authors mention on page 2 : left-sided Babinski sign, and more pronounced
DTRs on the left side”, however, those are uncommon in PSP. Were there any
other tests performed to exclude alternative diagnoses?

11) Was there any history of recent encephalitis, sensorial cortical deficits, alien
limb syndrome, history of stroke, neuroleptic use, cerebellar signs, or
hallucinations?

12) Did the patient have any family history of neurological diseases?

13) Was there any early sign of autonomic dysfunction?

14) What was the social history of the patient? Any drug use? What were his
occupation and ethnicity?

15) Was the patient followed up after discharge? How is he now?

16) Regarding the literature review on the neuroimaging of PSP, I advise the authors
to build a comparative table, including their patients clinical characteristics and
the main characteristics of the patients from the studies they included in their
review to improve readability of the text.

Author Response

Question/comment

Response

Reviewer 2

1) In the title, it is advisable that the authors include the study design type.

We have included the study design type (case presentation and short review).

2) It would be interesting if the authors could further describe the setting in which
the patient was seen by them. Was it an outpatient movement disorders clinic?
Public or private? Was it an academic setting?

We have added the missing information about our clinic.

3) Could the authors provide a video of the neurological exam of the patient,
perhaps labeling the PSP main signs?

Unfortunately, we do not have such a video.

4) On page 2 the authors mention “A brain MRI was also performed in 2019 but
was unremarkable (see Figures 1A and 2A).”However, figures 1 A and 2 A are
labeled with the “hummingbird” sign, “morning glory” sign, and “Mickey
mouse” sign. Was this perhaps a figure legend error? Can the authors provide
the MRI images from 2019?

The figure legends have been modified.

5) Can the authors further detail the initial pharmacological management of the
patient? Were any drugs tried before levodopa or with levodopa? Which were
the dosages, the levodopa regimen, and which type of formulation?

We have detailed the initial pharmacological treatment of the patient.

6) On page 2 the authors state: “ The patient repeated the brain MRI, which was
again unremarkable, apart from non-specific T2-high signal demyelinating
lesions involving the periventricular and subcortical white matter. “ Can the
MRI images be added to the paper?

Unfortunately, we do not have the images from that MRI.

7) Could the authors provide more details of the timeframe of the symptoms and
add months to their description? For example, in which month of 2022 was the
patient seen by them?

We have revised the information.

8) The authors mention in page 2 “who developed speech disturbance and walking
difficulties (accompanied by frequent backward falls), with insidious onset.”
Could the authors further clarify “insidious onset”? How long was that?

We have added the missing information.

9) It is important that the authors give further details on the patient’s clinical
history and physical exam. Could the authors further describe aspects such as
speech, mental exam, bulbar exam, oculomotor exam, limb and gait exam, if the
neurological symptoms were evaluated through any standardized scales besides
MMSE.

We have revised the information.

10) The authors mention on page 2 : “left-sided Babinski sign, and more pronounced
DTRs on the left side”, however, those are uncommon in PSP. Were there any
other tests performed to exclude alternative diagnoses?

We specified the paraclinical workup performed to exclude other possible causes.

11) Was there any history of recent encephalitis, sensorial cortical deficits, alien
limb syndrome, history of stroke, neuroleptic use, cerebellar signs, or
hallucinations?

We have added the missing information.

12) Did the patient have any family history of neurological diseases?

We have added the missing information.

13) Was there any early sign of autonomic dysfunction?

We have added the missing information.

14) What was the social history of the patient? Any drug use? What were his
occupation and ethnicity?

We have added the missing information.

15) Was the patient followed up after discharge? How is he now?

As of yet, the patient was not followed up after discharge.

16) Regarding the literature review on the neuroimaging of PSP, I advise the authors
to build a comparative table, including their patient’s clinical characteristics and
the main characteristics of the patients from the studies they included in their
review to improve readability of the text.

We do not have access to the main clinical characteristics of the patients from these studies. We only used the information to describe the imaging features seen in PSP.

Round 2

Reviewer 2 Report

Unfortunately, the authors were not able to sufficiently reply to the reviewer's questions.